# Advances in Mast Cell Activation by IL-1 and IL-33 in Sjögren’s Syndrome: Promising Inhibitory Effect of IL-37

**DOI:** 10.3390/ijms21124297

**Published:** 2020-06-16

**Authors:** Pio Conti, Luisa Stellin, Alesssandro Caraffa, Carla E. Gallenga, Rhiannon Ross, Spyros K. Kritas, Ilias Frydas, Ali Younes, Paolo Di Emidio, Gianpaolo Ronconi

**Affiliations:** 1Postgraduate Medical School, University of Chieti, 66013 Chieti, Italy; 2Department of Medicine and Science of Ageing, University of Chieti, 66013 Chieti, Italy; luisa.stellin@unich.it; 3School of Pharmacy, University of Camerino, 62032 Camerino, Italy; alecaraffa@libero.it; 4Department of Biomedical Sciences and Specialist Surgery, Section of Ophthalmology, University of Ferrara, 44121 Ferrara, Italy; gllcln@unife.it; 5University of Pennsylvania School of Veterinary Medicine, Philadelphia, PA 19104, USA; rhiross@upenn.edu; 6Department of Microbiology, University of Thessaloniki, 54124 Thessaloniki, Greece; skritas@vet.auth.gr; 7School of Veterinary Medicine, University of Thessaloniki, 54124 Thessaloniki, Greece; stavfd@hotmail.com; 8Centro Medico “Mai più Dolore”, 65100 Pescara, Italy; aliyounes@tiscali.it; 9Maxillofacial Surgery “G. azzini” Hospital, 64100 Teramo, Italy; paolodiemidio@gmail.com; 10Fondazione Policlinico Universitario A. Gemelli IRCCS, Università Cattolica del Sacro Cuore, 00100 Roma, Italy; gianpaolo.ronconi@policlinicogemelli.it

**Keywords:** mast cell, Sjögren syndrome, IL-1, IL-33, inflammation, cytokine, immunity

## Abstract

Sjögren’s syndrome (SS) is a chronic autoimmune inflammatory disease that affects primarily older women and is characterized by irreversible damage of the exocrine glands, including tear (xerophthalmia) and salivary glands (xerostomia). Secretory glands lose their functionality due to the infiltration of immune cells, which produce cytokines and cause inflammation. Primary SS is characterized by dry syndrome with or without systemic commitment in the absence of other pathologies. Secondary SS is accompanied by other autoimmune diseases with high activation of B lymphocytes and the production of autoantibodies, including the rheumatoid factor. Other cells, such as CD4^+^ T cells and mast cells (MCs), participate in SS inflammation. MCs are ubiquitous, but are primarily located close to blood vessels and nerves and can be activated early in autoimmune diseases to express a wide variety of cytokines and chemokines. In the SS acute phase, MCs react by generating chemical mediators of inflammation, tumor necrosis factor (TNF), and other pro-inflammatory cytokines such as interleukin (IL)-1 and IL-33. IL-33 is the specific ligand for ST2 capable of inducing some adaptive immunity TH2 cytokines but also has pro-inflammatory properties. IL-33 causes impressive pathological changes and inflammatory cell infiltration. IL-1 family members can have paracrine and autocrine effects by exacerbating autoimmune inflammation. IL-37 is an IL-1 family cytokine that binds IL-18Rα receptor and/or Toll-like Receptor (TLR)4, exerting an anti-inflammatory action. IL-37 is a natural inhibitor of innate and acquired immunity, and the level is abnormal in patients with autoimmune disorders. After TLR ligand activation, IL-37 mRNA is generated in the cytoplasm, with the production of pro-IL-37 and later mature IL-37 caspase-1 mediated; both precursor and mature IL-37 are biologically active. Here, we discuss, for the first time, the current knowledge of IL-37 in autoimmune disease SS and propose a new therapeutic role.

## 1. Introduction

The first description of autoimmune diseases occurred in the late 19th century. Today, there are over 80 autoimmune disorders, where it is noted that the immune system attacks its cells or tissues, causing functional damage to the organs [1]. About 7% of Americans were revealed to have an autoimmune disease, of whom women are more commonly affected. Normally, autoimmune diseases arise during adulthood and the cause is generally unknown [2].

A new mechanism for autoimmunity, which seems applicable in all systems, concerns organ involvement which can be inefficient as it can undergo cellular destruction, atrophy, and/or inhibition, but the true mechanism is still obscure. Infectious or chemical agents can induce autoimmunity through the mechanism of molecular mimicry that can occur when there is a similarity between the foreign antigen and the self-peptides. In this case, auto-reactive B or T cells are activated after exposure to foreign antigens leading to the autoimmune phenomenon.

Sjögren’s syndrome (SS) is an autoimmune chronic inflammatory disease, which can affect several organs and occur at any age, but primarily affects older women (90%) [3]. Primary Sjögren’s syndrome (pSS), characterized by lymphocytic infiltration of exocrine glands, anti-Ro/La autoantibody production, and hypergammaglobulinaemia, is of unknown pathogenesis, although it is attributed to reactive T cells, with a decrease in CD27^+^ memory B cells and an increase in circulating naïve CD27^−^ B cells. The percentage of pSS is not easy to establish, as estimates can vary according to different criteria for classifying patients, however using the criteria published by various authors, the prevalence of SS among women can vary between 0.6% and 2.1%.

The causes of SS are unknown, but this is an autoimmune disease in which genetic factors and viral infections can play a role in the development of the pathogenesis which causes irreversible damage to the exocrine glands [4]. Individuals with pSS experience tear gland dysfunctions with xerophthalmia, where the eye cannot produce tears, the salivary glands are also damaged, causing dry mouth due to the lack of saliva (xerostomia), and other systemic symptoms [5]. The secretory glands lose their functionality due to the infiltration of immune cells, including lymphocytes, which release inflammatory cytokines, activate other immune cells, and cause inflammation [6] (Figure 1). SS can be classified as primary, characterized by dry syndrome with or without systemic commitment and in the absence of other pathologies, or secondary, which is accompanied by other autoimmune diseases, such as systemic lupus erythematosus, scleroderma, rheumatoid arthritis, Hashimoto’s thyroiditis [7] (in this paper we mainly deal with pSS, however, the inflammatory phenomenon can occur in both the primary and secondary form of SS). In pSS, there is high activation of B lymphocytes with the production of autoantibodies such as anti-Ro60 and anti-La/SSB [8]. In this disease, the immune state appears mediated by T cells and excessive activation of B lymphocytes with the generation of autoantibodies such as anti-Ro/60 and anti-La/SSB, directed towards nuclear antigens. In addition, in the more common secondary SS form, associated with other rheumatic autoimmune diseases, there is an infiltration of immune cells in various organs, such as endocrine glands, and deposition of various autoantibodies including rheumatoid factor (RF), anti-Ro60/SS-A and anti-La/SS-B, which provoke inflammation mediated by pro-inflammatory cytokines [9]. Antibodies to Ro60 antigens are called anti-SSA/Ro and patients who express Ro60 antibodies in most cases are diagnosed with autoimmune conditions, including SS. Ro/La antigens are made up of 52Ro, 60Ro and La proteins, in addition to RNA particles. Anti-Ro/60 is the most common form in autoimmune diseases and is also associated with SS. Auto-antibodies damage the exocrine glands and induce inflammation caused by cytokines and chemokines [10]. The pathological aspect is assumed to have multifactorial etiology where genetic, hormonal, and environmental factors may play a relevant role. Generally, pSS seems to be associated with a genetic predisposition linked to HLA-B (HLA-Dw3, HLA-DRw52 and HLA-DR3) and individuals who have this genetic picture have a 20-fold greater risk of developing the disease than others [11].

Pathogenic viruses are potential biological elements that can trigger a number of diseases [12,13]. Infectious agents such as Epstein-Barr virus (EBV), cytomegalovirus, hepatitis C virus (HCV), and herpes virus can also damage the exocrine glands, resulting in low-grade systemic inflammation, thereby contributing to the development of pSS [14]. In pSS, EBV and human T-lymphotropic virus (HTLV-1) infection may be involved in promoting cross-reactivity of T or B cells with host antigens and autoantibody production [15]. Usually, in pSS the exocrine glands such as the salivary and tear glands are also infiltrated by CD4^+^ T lymphocytes. B cells play a crucial role in the pathogenesis of this disease, especially in subjects with hypergammaglobulinemia and the presence of RF, anti-SSA Ro, and antiSSB/La [16]. This pathological state can lead to non-Hodgkin lymphoma development which demonstrates the abundant development of B cells [17]. B cell activation, maturation, proliferation, and survival, are mediated by the B-cell activating factor (BAF) which is a protein of the TNF family [18]. In pSS, BAF levels are found to be increased both in serum and in salivary glands and can represent a therapeutic target [19]. Moreover, the acute phase of pSS disease is mainly mediated by TH1 and TH17 cells which favor the production of self-reactive inflammatory T cells [20]. Inflammation is fueled by the pro-inflammatory cytokines produced by CD4^+^ T lymphocytes that generate IL-1, IL-2, and interferon (IFN)-gamma that cause lumen stenosis, and damage to the glandular secretory ducts [21]. The mucosa and submucosa of the gastrointestinal tract are atrophic, with the infiltration of active plasma cells, and other immune cells [22].

Like other autoimmune diseases, in the pSS acute phase, innate immunity is particularly active, with dysregulation of pro-inflammatory cytokines/chemokines [23]. Endothelial cells constitute a physical and biological barrier, crucial for intravascular water/protein maintenance, but they are also a source of inflammatory cytokines [24]. In the acute phase, blood vessels, which include endothelial and activated mast cells (MCs), generate chemical mediators of inflammation and IL-1 family member cytokines such as IL-1 and IL-33 [25]. MCs are activated in autoimmunity, including in pSS, where they can provoke an inflammatory reaction in the whole organism that can go from low-grade to high-grade of inflammation [26]. IL-33 is involved in the polarization of T cells towards the Th2 cell phenotype and MC activation and plays a crucial role in inflammatory, infectious, and autoimmune diseases [27]. The IL-33/ST2 axis is crucial in experimental models of autoimmune diseases as ST2 deletion improves the development of T-cell-mediated disease [28]. When the production of IL-33 exceeds certain limits, there is an increase in systemic inflammation with worsening and exacerbation of autoimmune disorders. In an interesting article, it was reported that IL-33 is increased in autoimmune diseases, including pSS [29]. However, a prolonged increase of IL-33 can create dysfunction of both the FcεRI receptor and the inflammatory process [30]. Nevertheless, the inactivation of IL-33, the inhibition of its receptor, or using its soluble “decoy” receptor lead to important anti-inflammatory effects [31]. Hence, regulating the function of IL-33 produced by MCs could be crucial in the treatment of autoimmune inflammatory diseases mediated by this important cytokine [32].

IL-18 is another IL-1 family member that exerts pro-inflammatory and immune regulation in autoimmune diseases [33]. In allergic disorders mediated by MCs, an increase in IL-18 has been reported which causes stimulation and recruitment of MCs along with other cells of the myeloid lineage, increasing the inflammatory process in autoimmunity [34]. In the acute phase of pSS, where low-grade systemic inflammation can occur, there is an over-expression of several pro-inflammatory cytokines, such as TNF and IL-1 family members generated by immune cells including MCs [35]. It is well known that in pSS, as well as in other MC-mediated diseases, asthma, atopic dermatitis, multiple sclerosis, and rheumatoid arthritis, the blood levels of IL-33 are increased and its neutralization, the elimination of its receptor or the use of its soluble “decoy” receptor produce anti-inflammatory effects [1,36]. Patients may have rare but severe systemic symptoms, such as rash, fever, abdominal pain, or lung or kidney problems [37].

Therapeutic treatment of pSS uses non-specific drugs, although substantial progress in this field has been made in recent years [38]. The treatments are aimed at reducing inflammation of the glands, and some patients derive a limited benefit from the use of drugs that stimulate salivary and ocular secretions [39]. Hence, modulation of IL-33 produced by MCs may represent a promising strategy for the treatment of autoimmune diseases where cytokine dysregulation occurs [1]. The annoying and painful symptoms involving the exocrine glands in pSS are often treated with steroidal agents, analgesic drugs, immunosuppressants, and a chimeric monoclonal antibody that targets CD20 B cells, but the results are not satisfactory [40]. TNF stored in the granules and synthesized by MCs, is a pro-inflammatory cytokine that can be induced by IL-1 in different cell types, including MCs [41]. Today we know that the inhibition of TNF (with infliximab, etanercept, or monoclonal antibodies) and IL-1 (with anakinra a recombinant IL-1 receptor antagonist) in the treatment of autoimmune rheumatic diseases has given some positive but not completely satisfactory results [42]. In this article, we report that the pro-inflammatory cytokine IL-33 produced by IL-1 activated MCs can be an important therapeutic target. Therefore, blocking IL-33 with IL-37, a new inhibitory cytokine, results in an improvement of the low-grade of systemic inflammation, pain, and chronic fatigue syndrome, all symptoms present in pSS [43]. In fact, IL-37 is a powerful therapeutic tool that could also be administered alone or in combination with other therapies, for example, the one that foresees the use of anti-BAF receptors. IL-37 cytokine stimulates IL-10 and has an inhibitory effect on IL-6, a pro-inflammatory cytokine, but it is also involved in the differentiation of B cells, with increased levels in the serum, saliva, and tears in autoimmune rheumatic patients [43]. Therefore, IL-37 is a new inhibitory cytokine that could be relevant in the treatment of patients with pSS. Treatment with IL-37 may lead to a reduction in glandular inflammation, improving low-grade systemic inflammation. Cytokine IFN type I, also appears to be involved in pSS, an effect that can be inhibited with the administration of IL-37, although this statement needs to be confirmed [44].

Nowadays, there is no specific therapy for pSS, therefore this disease is commonly treated with a number of drugs, even with hydroxychloroquine an anti-rheumatic drug, but all with non-specific and unpleasant side effects. (For more information on IL-37, see hereunder the chapter dedicated to this cytokine).

## 2. Mast Cells (MCs)

MCs are immune cells hemopoietically derived, constitutively present in most tissues, particularly located close to blood vessels and nerves, and are best known for their role in IgE-associated allergic disorders [45]. MCs play a role as effector cells in anaphylactic reactions, mastocytosis, asthma, and cancer, and can also exacerbate autoimmune reactions [46]. They derive from CD34^+^ precursor cells which proliferate primarily in response to stem cell factor (SCF) and IL-6, and aid in microbial and parasitic infections [43]. MCs reside in the tissues, where they differentiate and mature locally and are widely distributed in all vascularized systems, close to blood vessels, nerves, smooth muscle cells, and glands [47]. There are two subtypes of MCs, MCα which are juvenile cells with a reduced number of granules and mature MCβ with a high number of granules. The functional difference between these two subtypes has yet to be established. The classic physiological function of MCs is to promote host resistance against bacterial or parasitic infections by limiting their devastating effects [48]. They can work according to the circumstances to perturb or help to restore homeostasis, contributing to disease or promoting health. MCs are “sentinels” of the body and reside close to surfaces exposed to the environment, such as the respiratory tract, skin, mucous membranes and glands where they react with pathogenic micro-organisms [49]. They are effector cells activated in immune responses and important initiators of innate immunity against pathogens, but they can also influence the adaptive immunity that contributes to pSS and other diseases (1) (Figure 2). MCs are relevant in helping the host to switch from innate to adaptive immunity and can be activated by different molecules without the intervention of IgE [50]. The activation of the classical MC pathway takes place with IgE which binds to its high-affinity receptor IgεRI (Kd = 10^−10^ M) and the reaction begins. Intracellular Ca^2+^ regulated by PLC and PKC occurs, resulting in the activation of IP3 and diacylglycerol (DAG), MAPK, ERK, JNK, and p38. These reactions activate the NF-κB with the secretion of pro-inflammatory cytokines/chemokines and arachidonic acid compounds. In fact, MCs regulate the migration and maturation of dendritic cells by producing cytokines IL-1 and TNF, along with highly inflammatory prostaglandin D2 [51,52]. Innate immune cells, including MCs, express toll-like receptor (TLR) TLR2 and TLR4. MCs can be also activated through the TLR2 and TLR4, and other cytokine receptors including IL-1R and ST2 [53].

Important experimental studies with great scientific progress have been made using MC-deficient c-Kit mutant mice, such as WBB6F1-KitW/KitW-v or C57BL/6-KitW-sh/KitW-sh [54]. Unfortunately, there are not many studies on humans and research carried out on rodents may not even reflect the effects in mankind.

Research has shown that MCs can influence cellular response in both B and T cells [55]. Antigen-specific T cells are pivotal in autoimmune diseases and can be recruited into inflamed tissues through the production of many compounds, including chemokines and adhesion molecules released by MCs [56]. Other factors produced by MCs that influence T cell migration are the chemokines: CXCL1, CCL2, CCL3, CCL4, CCL5, CCL20, CXCL10, IP-10; as well as some products of the arachidonic acid cascade, such as LTC4 (detected as leukotriene B4) and PGD2 [57]. Cooperation between T cells and B cells is important for the generation of antibodies, but the interaction between MCs and B cells is also crucial for the production of IgE mediated by IL-4 [58]. In addition, MCs produce a number of regulatory cytokines such as IL-4, IL-5, IL-6, and IL-13 which can also influence the generation of B cells [59].

pSS is characterized by an immune anomaly where there is an imbalance of self-reactive Th1/Th17 cells and B cells with activation of Th2 cells induced by different immune molecules, including IL-33, and production of several Th2 cytokines [60]. Therefore, it can be deduced that MCs can modulate antibody responses, and function both as immunoregulatory cells and as effector cells [61]. They also negatively intervene in autoimmune diseases by influencing a number of cells and tissues in an unclear way; an effect that appears reduced in mice genetically lacking MC [62]. Moreover, MC IgG1 activated by specific auto-immune antigens can cause the release of inflammatory mediators that contribute to aggravate autoimmune disease including pSS. Hence, they can play an effector function in innate immune responses, but they also contribute to acquired immunity in host defense and pSS [63]. All this is exercised through the production of inflammatory mediators released by MCs, influencing the activity of dendritic cells, T and B lymphocytes, and other immune and non-immune cells [64]. Therefore, MCs perform a pathogenic action in pSS, where they are activated and cross-talk using IL-1 and IL-33 with T and B cells that mediate the pathological state. Additionally, the production of IL-1 and TNF, and chemokines CXCL1 and CXCL2, recruit neutrophilic granulocytes that initiate the inflammatory lesion [65]. It is likely that by manipulating the function of MCs by suppressing inflammatory mediators such as IL-1 and IL-33, it will be possible to achieve a therapeutic improvement of pSS.

## 3. IL-1

Several studies reported that in almost all autoimmune diseases, innate immune cells play critical amplifying roles of the pathogenesis, and the importance of IL-1 family members in rheumatic disease has been highlighted [66].

The IL-1 family includes 11 members of which the first two, in order of discovery, are IL-1α and IL-1β respectively. In this study, we only consider IL-1β, the most studied, which will simply be called IL-1 [67]. IL-1 is a powerful pro-inflammatory cytokine important in autoimmune diseases, its inhibition is certainly helpful in reducing inflammation, pain, and the pathogenesis of the disease, with increased survival [43]. IL-1R is an IL-1 receptor and IL-1R3 is a co-receptor that binds IL-1, but also other cytokines such as IL-33, IL-36α, IL-36β, and IL-36γ, which are involved in inflammatory processes [67]. On the cell surface, IL-1 binds its specific IL1R receptor and IL-1R3 co-receptor or to TLR2 and TLR4, recruiting MyD88 which begins the kinase cascade called IRAK, followed by phosphorylation and activation of NF-κB [68]. After the formation of IL-1 mRNA, pro-IL-1 is generated, which through caspase-1 transforms it into secreted mature IL-1 [68]. The NLRP3 inflammasome, activated by pathogen-associated molecular patterns (PAMP) and Damage-associated molecular patterns (DAMP), plays an important role in inflammation and autoimmunity and participates in the regulation of secretion of pro-inflammatory cytokines [68]. The NLRP1 inflammasome, a regulator of the innate immune response, promotes caspase-1 which cleaves the inactive and immature pro–IL-1β to the mature bioactive IL-1β [69]. NLRP1 is genetically linked to the risk of contracting autoimmune diseases including pSS and other inflammatory diseases. In pSS, Th1 cytokines predominate in the infiltrated tissues of immune cells and IL-1 plays a crucial role [70]. Therefore, in pSS, several extra-glandular organs can be affected by inflammation with the development of pathologies such as interstitial nephritis, lung inflammation, liver injury, and cirrhosis, all diseases which involve MCs activated by IL-1, IL-33 and other immune proteins [71]. In pSS, glandular tissues such as salivary and lacrimal glands express the transcription of pro-inflammatory cytokines such as IL-1 and TNF produced by epithelial cells, B cells and MCs, showing that these cells are crucial [72].

Recently, in an important article, we reported that IL-33 stimulates MC to produce TNF, an effect increased in combination with the administration of a neurotransmitter, substance P, demonstrating the close relationship between IL-33 and TNF produced by activated MC and inflammation [73]. In addition, IL-33 induces vascular endothelial growth factor (VEGF) secretion from human MCs and is increased in autoimmune disorders, including psoriasis, contributing to the inflammatory state [74].

Activated MCs generate numerous cytokines that mediate autoimmune diseases, including pSS, and stimulate the release of inflammatory autocrine cytokines such as IL-33 and IL-1 [75]. In addition, stimulated human MCs producing TNF and IL-1 causes the synergistic secretion of IL-6, distinct from degranulation and contributing to the development of inflammation [76].

The use of agents that inhibit pro-inflammatory cytokines is very useful as they improve various diseases, including autoimmune disorders [77]. Specific anti-cytokine therapies would reduce the harmful effects that arise in autoimmune diseases including pSS. However, clinical treatments with existing inhibitors require frequent administration due to their short biological half-life in circulation. Therefore, in pSS, MCs producing IL-1 family members can aggravate the inflammatory state by activating other cytokines including IL-33 and IL-1, and their inhibition can be a significant aid.

## 4. IL-33

IL-33 (previously known as IL-1F11 or NF-HEV: nuclear factor from highly endothelial veins) is a member of the IL-1 family which is recognized as a specific ligand for the orphan member of the IL-1 ST2 receptor family [78]. The IL-1 superfamily includes ST2 which has its own unique IL-33 ligand and is elevated in several inflammatory diseases. The soluble form of ST2 acts as a decoy receptor and captures IL-33 and does not signal; while the cell membrane-linked form has biological activity through the activation of MyD88/NF–κB in several immune cells including MCs. Following the specific binding of IL-33 to the ST2 plasma membrane receptor, NF-κB and MAPK are induced [79]. IL-33 is secreted by smooth muscle cells, endothelial cells, macrophages, dendritic cells, fibroblasts, epithelial cells, and keratinocytes, all implicated in pSS [80]. IL-33 is involved in the proliferation of fibroblasts through processes dependent on the activation of kinases (MAPK, ERK, JNK, and p38), inducing the generation of [80] with infiltration of eosinophils. Moreover, IL-33 amplifies inflammatory responses and innate immunity by acting on the stimulation of TNF, stored and secreted by MCs [81]. MC ST2 level receptor is activated by IL-33 to produce pro-inflammatory cytokines such as IL-1, IL-6, and IL-13, as well as chemokines MCP-1 and MIP-1α [82]. IL-33 affects both innate and adaptive lymphoid cells and regulates the function of effector cells. IL-1 activates the MCs to produce IL-33 without degranulation, which has an autocrine action capable of reactivating MCs [76].

IL-33 is inactivated by caspase-1 and is mainly released in necrotic phenomena where it causes a rapid biological response and, for this reason, has been called pro-inflammatory “alarming” [43]. IL-33 is located in the nucleus, causing immune regulatory effects on homeostasis. When cleaved by proteases, more powerful mature forms are produced [83]. It performs the biological function by binding to the ST2 receptor which, with IL-1 AcP, induces MyD88, leading to the generation of other inflammatory cytokines. IL-33, which promotes signal transduction through IL-1R4, mediates inflammation in allergic and autoimmune responses by polarizing Th2 cells and activating IL-5 production in many cell types including MCs. In acute and chronic pSS, the expression of IL-33 is increased and its level may be associated with the severity of the disease. Under inflammatory conditions, abundant IL-33 is detected in human blood vessels [84].

Suppression of IL-33 leads also to the reduced production of TNF and IL-1 by various immune cells including MCs. These studies indicate that IL-33 exacerbates autoimmune pSS and mediates cytokine MC activation, and represents a new therapeutic approach for this disease.

## 5. IL-37

IL-37 is an anti-inflammatory cytokine that broadly suppresses innate and acquired immunity, and is abnormal in patients with autoimmune disorders [85]. It performs its anti-inflammatory functions by binding IL-18Rα receptor, also called IL-1R5, which binds IL-18, mediating inflammation [43]. Human IL-37 is a member of the IL-1 family that inhibits inflammation by reducing pro-inflammatory cytokine production. IL-37 (previously called IL-1F7) is a natural inhibitor of immunity and inflammation, of which five isoforms (a, b, c, d, e) have been identified, and cytokine “IL-37b” is the most studied and is discussed in the present study. IL-37 is generated by activated macrophages in response to TLR4 and/or IL-1R [68]. IL-37 synthesis and release occur after pro-inflammatory stimuli with activation of either IL-18Rα and/or TLR. After cell receptor activation, in the cytoplasm, IL-37 mRNA is generated which gives rise to the formation of pro-IL-37 which, through caspase-1, generates the mature IL-37 and moves (about 20%) to the nucleus [68] The cytoplasmic pro-IL-37 exits the cell, and biologically active IL-37 is generated through unknown protease(s). Both the pro-IL-37 and the mature IL-37 are biologically active.

Recent studies suggest that where IL-37 is decreased, there is an increase of inflammation and pro-inflammatory cytokine production [86]. Transgenic mice carrying human IL-37 genes demonstrated greater resistance to experimentally-induced inflammatory diseases. Several human cell types are capable of expressing cytokine IL-37 that maintains homeostasis by controlling the physiological immune status [68]. In normal condition, IL-37 is very low, but rises after an inflammatory stimulus, as occurs in autoimmune processes, as a defense reaction against inflammation [87]. IL-1 cytokine has two receptor binding sites IL-1RI and IL-1R AcP, one of which can be linked to IL-1Rα preventing IL-1β binding and carrying out anti-inflammatory activity. The inhibition of IL-1, IL-33, and TNF secreted by MCs is very important in pSS, but other pro-inflammatory cytokines can contribute to the disease. However, if the suppression is too strong, it would lead to an increased risk of opportunistic infections, including tuberculosis [88] (Figure 3). IL-37 suppresses innate immune responses by suppressing mTOR and acts by inhibiting acquired responses [68]. The inhibitory effect of IL-37 is exerted on pro-inflammatory cytokines of the IL-1 family members [89]. On the other hand, it stimulates IL-10 which also exerts anti-inflammatory activity. In autoimmune diseases, IL-37 and IL-10 contribute to the tolerance process by suppressing MHC class II and inflammation [90]. In experimental autoimmune rheumatic diseases on mice, the administration of human IL-37 inhibits inflammation, with a reduction of cytokines IL-1β, IL-6, TNF, and chemokine CCL2 [43], all products of MCs (Figure 4). IL-37 increases in autoimmune rheumatic diseases, which is related to the inflammatory state involving T cell activation.

Here, our studies underline for the first time the importance of IL-1 and IL-33 secreted by MC activation in pSS and support the efficacy of IL-37 on suppression of inflammation. In addition, the inhibitory effect of IL-37 could be a new therapeutic strategy in the treatment of inflammatory pSS autoimmune diseases.

As IL-37 is an anti-inflammatory, but also an immunosuppressive cytokine, its application should be carried out very carefully.

## Figures and Tables

**Figure 1 ijms-21-04297-f001:**
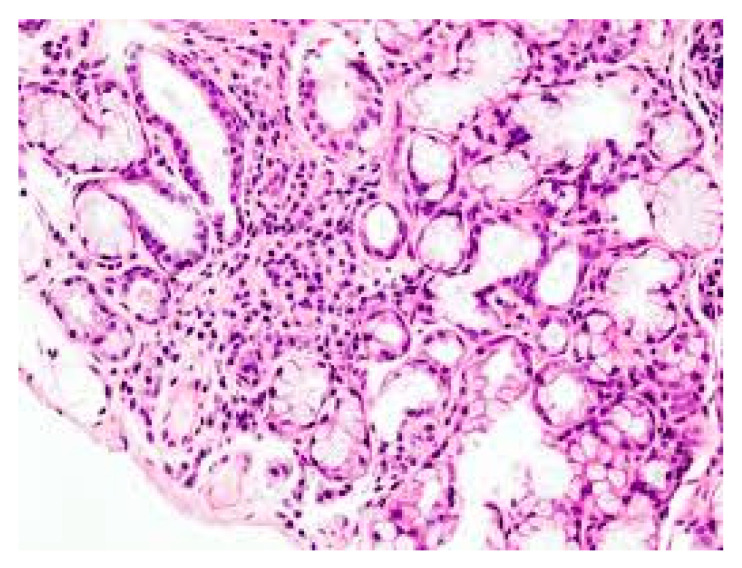
Lymphocyte infiltration in primary Sjogren’s syndrome (pSS) xerostomia (biopsy). (The tissue was coloured with Toleudine blue 0.1% then was analysed under optical miscroscopy magnification ×10).

**Figure 2 ijms-21-04297-f002:**
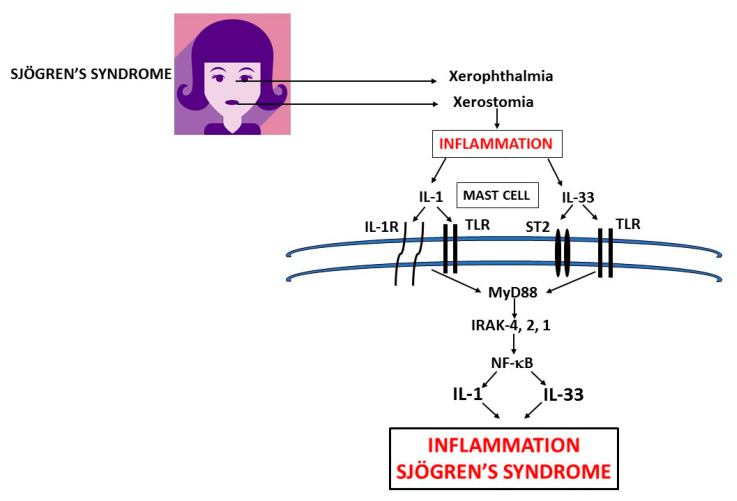
Xerophthalmia and xerostomia glands in primary Sjogren’s syndrome (pSS) inflammation release pro-inflammatory cytokines IL-1 and IL-33 which activate mast cells (MCs) to produce IL-1 family members exacerbating inflammation in pSS. In this figure, we show the biosynthesis of IL-1 and IL-33 which are mediators of systemic inflammation in primary and secondary Sjogren’s syndrome. IL: interleukin; TLR: Toll-like receptor; ST: soluble transmembrane; MyD: Myeloid differentiation primary response; IRAK: IL-1 receptor-associated kinase; NF-κB: nuclear factor kappa-light-chain-enhancer of activated B cells.

**Figure 3 ijms-21-04297-f003:**
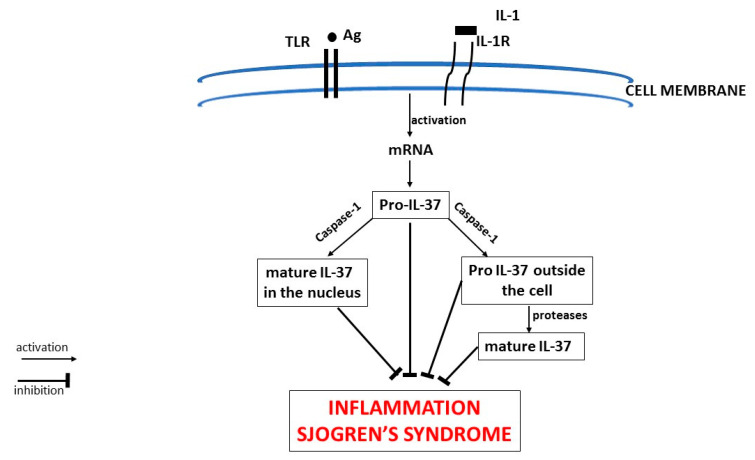
The antigenic activation of Toll-like receptor (TLR) and/or interleukin (IL)-1R receptor with IL-1 leads to the generation of IL-1 mRNA with the production of pro-IL-37, which by means of a caspase-1 causes the release of IL-37 mature in the nucleus. Outside the cell, pro-IL-37 reacts with an unknown protease to produce the mature IL-37. Both, pro-IL-37 and the mature form are active in inhibiting inflammation in Sjogren’s syndrome. Here we confirm the anti-inflammatory activity of IL-37 in several diseases, including Sjogren’s syndrome. TLR: Toll-like receptor; IL: interleukin; mRNA: messenger ribonucleic acid; Ag: antigen.

**Figure 4 ijms-21-04297-f004:**
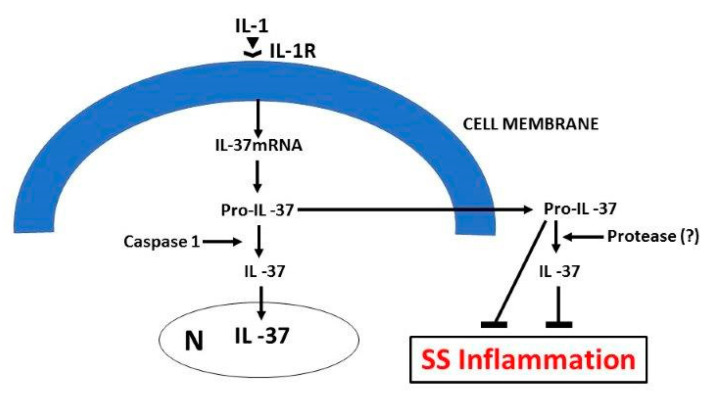
In this figure, we show the mechanism of IL-37 exerting immuno-inhibitory function in Sjogren’s syndrome inflammation. IL: interleukin; mRNA: messenger ribonucleic acid.

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
