# Peer review of "Advances in Mast Cell Activation by IL-1 and IL-33 in Sjögren’s Syndrome: Promising Inhibitory Effect of IL-37"

_ijms, 2020, doi:10.3390/ijms21124297_

Round 1

Reviewer 1 Report

Sjögren's syndrome (SS) is a chronic autoimmune inflammatory disease characterized xerophthalmia and xerostomia, autoantibody production and cellular infiltration of salivary and lacrimal glands. Mast cells generate chemical mediators of inflammation, tumor necrosis factor (TNF) and other pro-inflammatory cytokines such as interleukin (IL)-1 and IL-33. IL-37, an IL-1 family cytokine, exerts an anti-inflammatory action. In this article the authors describe that the pro-inflammatory cytokine IL-33 produced by IL-1 activated mast cells can be an important therapeutic target. They also suggest the cautious use of IL-37 for suppressing inflammation.

This is a useful review regarding the role of IL-1, IL-33, and IL-37 in Sjogren’s syndrome. There are some concerns to be addressed.

It would be of used to use sub-sections for each section for ease of reading

e.g 1. Introduction

  1. a) Autoimmunity

Here, the authors could discuss the current theories of how autoimmunity can occur

e.g. molecular mimicry, failure in deletion of autoreactive cells…

  1. b) Sjogren’s syndrome

What percentage of autoimmunity does SS comprise. Characteristics of SS and pSS

pSS and sSS

  1. c) Autoantibodies in SS
  2. d) Cytokines and chemokines in SS
  3. e) Cellular aspects of SS
  4. f) Therapeutic aspect of SS

Similarly for other sections in the paper:

The title describes IL1 and IL33, but abstract does not mention much about IL 33, but mentions mainly IL 37.

"antibodies such as anti-Ro/SSA and anti-La/SSB (8). In this disease, the immune state appears"

It should be Ro60

"antibodies, such as rheumatoid factor, anti-SS-A and anti-SS-A/Ro, directed towards nuclear"

This should be “antibodies, such as rheumatoid factor, anti-Ro60/SS-A and anti-La/SS-B, directed towards nuclear

It will be useful to add a small paragraph regarding Ro60/SSA, and La/SSB autoantigen.

Line 65: In addition, in the more common secondary SS form, associated with other rheumatic autoimmune diseases, there is an infiltration of immune cells and deposition of various auto-antibodies including rheumatoid factor (RF), anti-SS-A/Ro and anti-SS-B/La, which provoke inflammation mediated by pro-inflammatory cytokines (9).

Which organs do the immune cells infiltrate in sSS? Where do the autoantibodies deposit?

Need to specify that when the authors say SS it is primary SS.

Need to specify when the authors say anti-Ro/SSA, it is anti-Ro60.

Abbreviations need to expanded when first used, e.g. SSA, SSB. HCV, etc.

More information regarding the IL-33/ST2 axis would be useful

Line 108: is it the epsilon symbol that was meant to be given in FcRI receptor? Again in line 168 the symbol used, is it for epsilon?

Again the same symbol used in Line 221

Line 144-Treatment with IL-37 may lead to a reduction in glandular inflammation and an increase in secretion, improving the low-grade of systemic inflammation.

Treatment with IL-37 lead to increase in all kinds of secretion?

Which TLR activate mast cells?

Line 208: …influencing the activity of dendritic cells, T and B lymphocytes and other immune and not immune cells (64).

Non-immune cells?

Line 225: On the cell surface, IL-1 binds its specific IL1R receptor and IL-1R3 co-receptor or to TLR

It is not clear which specific TLR the authors are referring to here?

Recent studies suggest that where IL-37 is decreased, there is an increasd of inflammation and pro-inflammatory cytokine production (86).

This sentence needs to be rewritten- “is increased inflammation” instead of “is an increasd of inflammation”

Line 332: Since IL-37 is a new cytokine that appears in the immune field

This part of the sentence needs to be rewritten. The cytokine does not “appear” in the “immune field”

Author Response

REPLIES TO REVIEWER No. 1

We thank this reviewer for the positive comments to our paper and for his suggestions to improve the manuscript.

We added the percentage of the SS autoimmunity and we wrote that we mostly deal with primary SS in the entire paper.

We added further information regarding IL-33 in the abstract.

We corrected anti Ro/SSA in anti-Ro60, as suggested, throughout the paper

We also corrected Anti-SS-A and anti-SS-A/R,o as suggested

We added the organs that are infiltrated with immune cells in sSS, as well as where the antibodies are deposited.

Throughout the paper we specified whether it was primary SS.

HCV has been written in full – hepatitus C virus when first used, as well as the other acronyms in the paper.

Again, the Greek symbol has been corrected in line 222.

The sentence regarding treatment with IL-37 has been deleted as it may cause confusion.

We report that TLR2 and TLR4 activate mast cells. Thank you for pointing out this lack.

The typo not immune cells has been corrected in non-immune cells.

As reported above, the TLRs involved in inflammation are mostly TLR 2 and 4.

This sentence (Line 86) has been re-written and it is now clearer.

Also in Line 33 the sentence has been re-written.

Thank you for your suggestions.

Reviewer 2 Report

In this paper, Conti and colleagues reviewed the role of mast cells, IL-1, IL-33 as well as IL-37 in the pathogenesis of Sjögren’s syndrome. It highlighted the immuno-inhibitory function of IL-37, a natural of innate and acquired
immunity and proposed a new therapeutic role. Overall, the review article is well organized and presented. Following concerns should be addressed.

  • Figures 2, 3 are general graphs describing the signaling pathways activated by IL-1, IL-33 or leading to IL-37 production, not in the context of Sjögren’s syndrome. The authors are encouraged to depict SS specific pathogenesis or biological processes. 
  • In the title, the author attempted to emphasize the role of IL-37 in SS, which is also the novelty of the review. It would be informative to draw a figure on how IL-37 exerts immuno-inhibitory functions on both innate and adaptive immunity. 
  • Typos in the manuscript need to be checked. 

Author Response

REVIEWER No. 2

We thank this reviewer for the positive comments regarding our article.

In Figs. 2 and 3 we clarified that the inflammatory and non-inflammatory cytokine production are invovled in SS. In addition, we specified the connection between cytokines and SS.

As the referee suggested, we depict another figure regarding the biosynthesis of IL-37 and its inhibitory effect on SS.

Round 2

Reviewer 2 Report

The authors have addressed my concerns in the revised manuscript.